# Frequency Guidance Matters: Skeletal Action Recognition by Frequency-Aware Mixed Transformer

Wenhan Wu
University of North Carolina at Charlotte
Department of Computer Science
Charlotte, NC, USA
wwu25@uncc.edu

Ce Zheng
Carnegie Mellon University
Robotics Institute
Pittsburgh, PA, USA
cezheng@andrew.cmu.edu

Zihao Yang
Microsoft Corporation
One Inventory Organization
Redmond, WA, USA
mattyang@microsoft.com

Chen Chen
University of Central Florida
Center for Research in Computer Vision
Orlando, Florida, USA
chen.chen@crcv.ucf.edu

Srijan Das
University of North Carolina at Charlotte
Department of Computer Science
Charlotte, NC, USA
sdas24@uncc.edu

Aidong Lu
University of North Carolina at Charlotte
Department of Computer Science
Charlotte, NC, USA
aidong.lu@uncc.edu

## Abstract

Recently, transformers have demonstrated great potential for modeling long-term dependencies from skeleton sequences and thereby gained ever-increasing attention in skeleton action recognition. However, the existing transformer-based approaches heavily rely on the naive attention mechanism for capturing the spatiotemporal features, which falls short in learning discriminative representations that exhibit similar motion patterns. To address this challenge, we introduce the **Freq**uency-aware **Mix**ed Trans**former** (FreqMixFormer), specifically designed for recognizing similar skeletal actions with subtle discriminative motions. First, we introduce a frequency-aware attention module to unweave skeleton frequency representations by embedding joint features into frequency attention maps, aiming to distinguish the discriminative movements based on their frequency coefficients. Subsequently, we develop a mixed transformer architecture to incorporate spatial features with frequency features to model the comprehensive frequency-spatial patterns. Additionally, a temporal transformer is proposed to extract the global correlations across frames. Extensive experiments show that FreqMiXFormer outperforms SOTA on 3 popular skeleton action recognition datasets, including NTU RGB+D, NTU RGB+D 120, and NW-UCLA datasets. Our project is publicly available at: https://github.com/wenhanwu95/FreqMixFormer.

## CCS Concepts

• **Computing methodologies** → **Artificial intelligence**; **Computer vision**.

## Keywords

Skeleton Action Recognition, Frequency, Transformer

**ACM Reference Format:**

Wenhan Wu, Ce Zheng, Zihao Yang, Chen Chen, Srijan Das, and Aidong Lu. 2024. Frequency Guidance Matters: Skeletal Action Recognition by Frequency-Aware Mixed Transformer. In *Proceedings of Proceedings of the 32nd ACM International Conference on Multimedia (MM'24)*. ACM, New York, NY, USA, 10 pages. https://doi.org/10.1145/3664647.3681009

## 1 Introduction

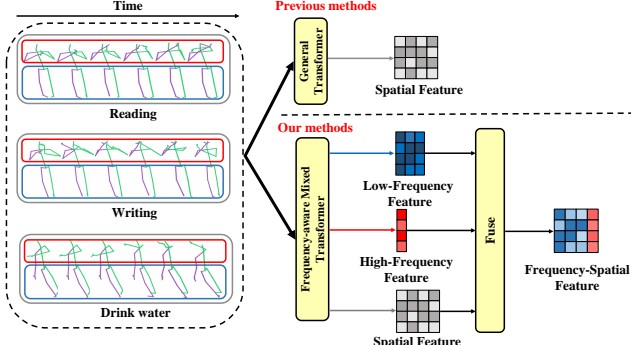

**Figure 1: The overall design of our Frequency-aware Mixed Transformer. Our FreqMixFormer model overcomes the limitations of traditional transformer-based methods, which cannot effectively recognize confusing actions such as reading and writing due to the straightforward process of skeleton sequences. As highlighted with the colored boxes, the FreqMixFormer introduces the frequency domain and extracts high-frequency features, which often indicate subtle and dynamic movements (red), and low-frequency features, which are associated with slow and steady movements (blue). These features are then fused with spatial features. Our results demonstrate that the integrated frequency-spatial features significantly improve the model's capability to discern discriminative joint correlations.**

Human action recognition, a vital research topic in computer vision, is widely applied in various applications, including visual surveillance [48, 49], human-computer interaction [18, 37], and autonomous driving systems [28, 30]. Particularly, skeleton sequences represent the motion trajectories of human body joints to characterize distinctive human movements with 3D structural pose information, which is robust to surface textures and backgrounds. Consequently, skeletal action recognition stands out as an effective approach for recognizing human actions compared to RGB-based [52, 65] or Depth-based [61, 77] methods. Early skeleton-based works typically represent the human skeleton as a sequence of 3D joint coordinates or a pseudo-image, then adapted Convolutional Neural Networks (CNNs) [15, 19, 20] or Recurrent Neural Networks (RNNs) [12, 21, 68] to model spatial features among joints. However, unlike static images, skeleton data embodies dynamic and complex spatiotemporal topological correlations that CNNs and RNNs fail to capture. Therefore, to effectively model the skeletal information encapsulated within the topological graph structure that reflects human anatomy, Graph Convolutional Networks (GCNs) [8–10, 22, 23, 25, 46, 63, 66] are utilized. Nevertheless, as graph information progresses through deeper layers, the model may lose vital joint correlations and direct propagation, diminishing its ability to capture long-range interactions between distant frames.

Recently, Transformer [53] has acquired promising results in human action recognition in various data modalities such as RGB [29, 44], depth [57, 58], point cloud [33, 60], and skeleton [10, 56]. The Transformer's ability to model the intrinsic representation of human joints and sequential frame correlations makes it a suitable backbone for skeleton-based action recognition. Despite the notable achievements of several transformer-based studies [4, 14, 31, 41, 47, 62, 64, 70, 76], they have yet to surpass the accuracy benchmarks set by GCNs. We hypothesize that the primary issue contributing to this gap is - GCNs, through their localized graph convolutions, effectively capture the human spatial configuration essential for action recognition. In contrast, traditional transformers, utilizing global self-attention operations, lack the inductive bias to inherently grasp the skeleton's topology. Although these global operations model overall motion patterns within a skeleton sequence, the self-attention mechanism in transformers may dilute the subtle local interactions among joints. Additionally, as attention scores are normalized across the entire sequence, subtle yet crucial discrimination in action sequences might be ignored if they do not substantially impact the overall attention landscape.

To bridge this gap, we focus on **improving transformers' capability to learn discriminative representation of subtle motion patterns.** In this work, we aim to aggregate the frequency-spatial features by introducing Discrete Cosine Transform (DCT) [1] into a mixed attention transformer framework [11, 64] to encode joint correlations in the frequency domain to explore the frequency-based joint representations. The motivation behind the representations is straightforward and intuitive: the frequency components can represent the entire joint sequence [36] and are sensitive to subtle movements. As a result, we introduce a novel Frequency-aware Mixed Transformer (FreqMixFormer) for skeleton action recognition to capture the discriminative correlations among joints. The key steps of our approach are outlined as follows:

*Firstly*, we formulate a Frequency-aware Attention module for transferring spatial joint information to the frequency domain, where the skeletal movement dependencies (similarity score between Queries $Q$ and Keys $K$) are embedded in the spectrum domain with a distinct representation based on their energy. As illustrated in Fig.1, discriminative skeleton features with similar patterns (e.g., confusing actions like reading and writing) can be effectively learned by leveraging their physical correlations. In the context of these skeleton sequences, minor movements that contain subtle variations and exhibit rapid spatial changes are effectively compressed into **high-frequency components** with lower energy (highlighted with the red box). Conversely, actions that constitute a larger portion of the sequence and change slowly over time in the temporal domain are compressed into **low-frequency components** with higher energy (shown in the blue box). Subsequently, a frequency operator is applied to accentuate the high-frequency coefficients while diminishing the low-frequency coefficients, thereby enabling selective amplification and attenuation for fine-tuning within the frequency domain. *Secondly*, we propose a transformer-based model that utilizes mixed attention mechanism to extract spatial and frequency features separately with self-attention (SA) and cross-attention (CA) operations, where SA and CA extract joint dependencies and contextual joint correlations respectively. An integration module subsequently fuses the features from both the frequency and spatial domains, resulting in frequency-spatial features. These features are then fed into a temporal transformer, which globally learns the inter-frame joint correlations (e.g., from the first to the last frame), effectively capturing the discriminating frequency-spatial features temporally. Our contributions are summarized as follows:

- We propose a **Frequency-aware Attention Block** (FAB) to investigate frequency features within skeletal sequences. A frequency operator is specifically designed to improve the learning of frequency coefficients, thereby enhancing the ability to capture discriminative correlations among joints.
- Consequently, we introduce the **Frequency-aware Mixed Transformer** (FreqMixFormer) to extract frequency-spatial joint correlations. The model incorporates a temporal transformer designed to enhance its ability to capture temporal features across frames.
- Our proposed FreqMixFormer outperforms state-of-the-art performance on three benchmarks, including NTU RGB+D [45], NTU RGB+D 120 [32], and Northwestern-UCLA [54].

## 2  Related Work

**Frequency Representation Learning for Skeleton-based Tasks.** Traditional pose-based methods aim to extract motion patterns directly from the poses for trajectory-prediction [27, 50], pose estimation [6, 74], action recognition [10, 66]. The representations derived from pose space naturally reflect physical characteristics (spatial dependency of structure information) and motion patterns (temporal dependency of motion information), making it challenging to encode poses in a spatiotemporal way. Motivated by a strong ability to encode temporal information in the frequency domain smoothly and compactly [3], several recent works [13, 26, 36, 55] utilize discrete cosine transform (DCT) to convert the temporal

motion to frequency domain for frequency-specific representation learning.

In skeleton action recognition, only a few works [5, 7, 16, 42] have considered frequency representations so far. [16] proposed a multi-feature branches framework to extract subtle frequency features with fast Fourier transform (FFT) and spatial-temporal joint dependencies, aiming to build a multi-task framework in skeleton action recognition. [7] adopts discrete wavelet transform with a GCN-based decoupling framework to decouple salient and subtle motion features, aiming for fine-grained skeleton action recognition. While our interest aligns with frequency-based modeling, we opt for a DCT-based approach since its frequency coefficients are well-distributed in the frequency domain, benefiting the discriminative motion representation learning.

**Transformer-based Skeleton Action Recognition.** Many recent works adopt transformers for human pose estimation [71, 73] and skeleton action recognition [35, 43] to explore joint correlations via attention mechanism. ST-TR [41] is the first to introduce the transformer to process skeleton data with spatial transformer and temporal transformer, proving its effectiveness in action recognition. Many follow-up works [14, 35, 43, 64, 70] keep employing this spatial-temporal structure for skeleton recognition with different configurations. STTFormer [43] proposed a tuple self-attention mechanism for capturing the joint relationships among frames. FG-STFormer [14] was developed to understand the connections between local joints and contextual global information across spatial and temporal dimensions. SkeMixFormer [64] introduced mixed attention method [11] and channel grouping techniques into spatiotemporal structure, enabling the model to learn the dynamic multivariate topological relationships. Besides these methods that focus on model configurations, [38] designed a partitioning strategy with the self-attention mechanism to learn the semantic representations of the interactive body parts. [31] presented an efficient transformer with a temporal partitioning aggregation strategy and topology-aware spatial correlation modeling module.

Most of the transformer-based methods mentioned above mainly focus on configuration improvement and spatiotemporal correlation learning without exploiting the skeletal motion patterns in the frequency domain. In this work, we propose a frequency-based transformer with a frequency-spatial mixed attention mechanism, leveraging joint representation learning.

## 3 Methodology

### 3.1 Preliminaries

**Transformer.** Self-Attention is the core mechanism of the transformer [53]. Given the input $X \in \mathbb{R}^{C \times D}$, where $C$ is the number of patches and $D$ is the embedding dimension, $X$ is first mapped to three matrices: *Query* matrix $Q$, *Key* matrix $K$ and *Value* matrix $V$ by three linear transformation:

$$Q = XW_Q, \quad K = XW_K, \quad V = XW_V \tag{1}$$

where $W_Q$, $W_K$ and $W_V \in \mathbb{R}^{D \times D}$ are the learnable weight matrices.

The self-attention score can be described as the following mapping function:

$$Attention(Q, K, V) = Softmax(\frac{QK^\top}{\sqrt{d}})V \tag{2}$$

where $QK^\top$ is the similarity score, $\frac{1}{\sqrt{d}}$ is the scaling factor that prevents the softmax function from entering regions where gradients are too small. Next, the Multi-Head Self-Attention (MHSA) function is introduced to process information from different representation subspaces in different positions. The MHSA score is expressed as:

$$MHSA(Q, K, V) = Concat(H_1, H_2, \ldots, H_h)W_{out} \tag{3}$$

where $H_i = Attention(Q_i, K_i, V_i)$, $i \in \{1, 2, \ldots, h\}$ is the single attention head, $W_{out}$ is a linear projection $\in \mathbb{R}^{D \times D}$.

**Baseline.** The existing transformer-based skeleton action recognition methods rely heavily on plain self-attention blocks mentioned above to capture spatiotemporal correlations, ignoring the contextual information among different blocks. Thus, we simply adopt off-the-shelf SkeMixformer [64] as our baseline for capturing spatial skeletal features, where the contextual information can be extracted based on a mixed way: 1) Cross-attention, an asymmetric attention scheme of mixing *Query* matrix $Q$ and *Key* matrix $K$, leveraging asymmetric information integration. 2) Channel grouping, a strategy that divides the input into unit groups to capture multivariate interaction characteristics, preserving the inherent features of the skeleton data by avoiding the full self-attention's dependency on global complete channels.

However, SkeMixFormer falls short of modeling discriminative motion patterns, thereby not fully leveraging its representational potential. In light of the baseline's limitations, we introduce our proposed FreqMixFormer to verify the effectiveness of frequency-spatial features over purely spatial ones. The detailed components of our model are elaborated in the following sections.

### 3.2 Overview of FreqMixFormer

The overall architecture of FreqMixFormer is illustrated in Fig. 2. Given the input $X \in \mathbb{R}^{J \times C \times F}$ is embedded by joint and positional embedding layers to represent a skeleton sequence with a consistent frame count of $F$, where $C$ denotes the dimensionality of the joint, and $J$ represents the number of joints in each frame. Then a partition block is proposed for capturing multivariate interaction association characteristics, where $X$ is divided into $n$ unit groups ($n = 3$ in Fig. 2 for example) by channel splitting to facilitate interpretable learning of joint adjacency. The split unit is expressed as $x_i \in \mathbb{R}^{J \times (C/n) \times F}$ and $X \leftarrow Concat[x_1, x_2, \ldots, x_i]$, where $i = 1, 2, \ldots, n$. Next, we feed unit inputs $x_i$ to the Frequency-aware Mixed Transformer based on self-attention and cross-attention mechanisms among Spatial Attention Blocks and Frequency-aware Attention Blocks. Afterward, frequency-spatial mixed features are processed with a Temporal Attention Block to learn inter-frame correlations. The final outputs are further reshaped and passed to an FC layer for classification.

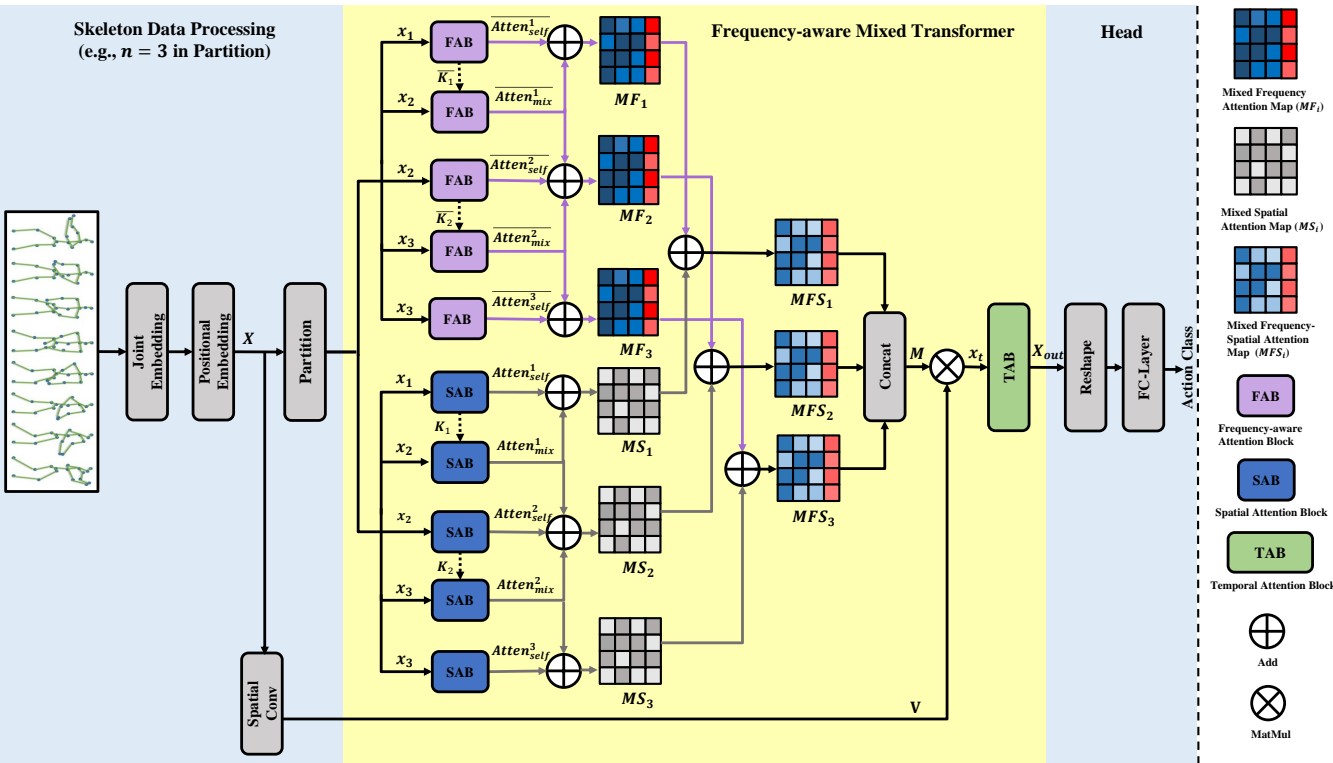

**Figure 2: Overview of the proposed FreqMixFormer. Given the skeleton sequence, we first perform the joint and positional embedding to get the embedded $X$. Then $X$ is divided into $n$ ($n = 3$ as an example in this figure) unit groups as the input $x_i$. The explanation of data partition is available in Section 3.2. Next, $x_i$ passes through the Frequency-aware Mixed Transformer to extract the mixed frequency-spatial attention maps $MFS_i$ (the definition is available in Section 3.4 ), which contain the joint fusion patterns from the frequency and spatial domains. These maps are subsequently concatenated into a feature $M$ along with the *Value* $V$ as the input of the temporal attention block, leading to an inter-frame joint correlation learning, and the corresponding output $X_{out}$ is passed to an FC-layer for the classification.**

## 3.3 Discrete Cosine Transform (DCT) for Joint Sequence Encoding

Let $x \in \mathbb{R}^{J \times C \times F}$ denotes the input joint sequence, the trajectory of the $j$-th joint across $F$ frames is denoted as $X_j = (x_{j,0}, x_{j,1}, ..., x_{j,F})$. While existing transformer-based skeleton action recognition methods only use this $X_j$ as an input sequence for skeletal correlation representation learning in the spatial domain, we propose to adopt a frequency representation based on Discrete Cosine Transform (DCT). Different from the previous DCT-based trajectory representation learning methods [25, 36, 72], which discard some of the high-frequency coefficients for providing a more compact representation, we not only keep all the DCT coefficients but also enhance the high-frequency parts and reduce the low-frequency parts. **The main motivations behind this are:** (i) High-frequency DCT components are more sensitive to those subtle discrepancies that are difficult to discriminate in the spatial domain (e.g., the hand movements in reading and writing, which are illustrated in Fig. 1). (ii) Low-frequency DCT coefficients reflect the movements with steady or static motion patterns, which are not discriminative enough in recognition (e.g., the lower body movements in reading and writing, which are also illustrated in Fig. 1). (iii) The cosine transform exhibits excellent energy compaction properties to concentrate the majority of the energy (low-frequency coefficients) into the first few

coefficients of the transformation, meaning it is well-distributed for amplifying subtle motion features.

Thus, we apply DCT to each trajectory individually. For trajectory $X_j$, the $i$-th DCT coefficient is calculated as:

$$C_{j,i} = \sqrt{\frac{2}{F}} \sum_{f=1}^{F} x_{j,f} \frac{1}{\sqrt{1 + \delta_{i1}}} \cos \left[ \frac{\pi(2f - 1)(i - 1)}{2F} \right] \quad (4)$$

where the *Kronecker* $\delta_{ij} = 1$ if $i = j$, otherwise $\delta_{ij} = 0$. In particular, $i \in \{1, 2, \ldots, F\}$, the larger $i$, the higher frequency coefficient. These coefficients enable us to represent skeleton motion within the frequency domain effectively. Besides, the original input sequence in the time domain can be restored using Inverse Discrete Cosine Transform (IDCT), which is given by:

$$x_{j,f} = \sqrt{\frac{2}{F}} \sum_{i=1}^{F} C_{j,i} \frac{1}{\sqrt{1 + \delta_{i1}}} \cos \left[ \frac{\pi(2f - 1)(i - 1)}{2F} \right] \quad (5)$$

where $j \in \{1, 2, \ldots, F\}$.

To use DCT coefficients in the transformer, we further introduce a Frequency-aware Mixed Transformer for extracting mixed frequency-spatial features in the next section.

## 3.4 Frequency-aware Mixed Transformer

**Mixed Spatial Attention.** Given a split input $x_i \in \mathbb{R}^{J \times (C/n) \times F}$ mentioned in Section 3.2, the basic *Query* matrix and *Key* matrix for each sequence are extracted along the spatial dimension:

$$Q_i, K_i = ReLU(linear(AvgPool(x_i))), \qquad (6)$$

where $i = 1, 2, \ldots, n$. In Eq. 6, *AvgPool* denotes adaptive average pooling for smoothing the joint weight and minimizing the impact of noisy or less relevant variations within the skeletal data, and an FC-layer with a ReLU activation operation is applied to ensure $Q_i$ and $K_i$ are globally integrated. Then, the self-attention is expressed as:

$$Atten^i_{self} = Softmax(\frac{Q_i K_i^\top}{\sqrt{d}}) \qquad (7)$$

In order to enable richer contextual integration across different unit groups, inspired by [11], a cross-attention strategy is proposed, where $K_i$ is shared between adjacent attention blocks. The cross attention is expressed as:

$$Atten^i_{mix} = Softmax(\frac{Q_{i+1} K_i^\top}{\sqrt{d}}) \qquad (8)$$

Each mixed attention map is formulated as:

$$MS_i = Atten^i_{self} + Atten^i_{mix} + Atten^{i-1}_{mix} \qquad (9)$$

where the number of this association mixed-attention maps is based on the number of unit groups (e.g., $n = 3$ in Fig. 2). These mixed-attention maps are extracted by several SABs (Spatial Attention Blocks, illustrated in Fig. 3 (a)) for spatial representation learning.

**Mixed Frequency-Spatial Attention.** We apply DCT to obtain the corresponding frequency coefficients from the split joint sequence $x_i$, and then the inputs to FABs (Frequency-aware Attention Blocks, see in Fig. 3) can be denoted as $DCT(x_i)$, where $DCT(\cdot)$ denotes the transform expressed in Eq. 4. Similar to the mixed spatial attention, we obtain the *Query* and *Key* values along the frequency domain:

$$\overline{Q}_i, \overline{K}_i = ReLU(linear(AvgPool(DCT(x_i)))) \qquad (10)$$

The corresponding frequency-based self-attention and mixed-attention maps are:

$$\overline{Atten}^i_{self} = Softmax(\frac{\overline{Q}_i \overline{K}_i^\top}{\sqrt{d}}) \qquad (11)$$

$$\overline{Atten}^i_{mix} = Softmax(\frac{\overline{Q}_{i+1} \overline{K}_i^\top}{\sqrt{d}}), \qquad (12)$$

Thus, the mixed frequency attention maps are expressed as:

$$\overline{MF_i} = \overline{Atten}^i_{self} + \overline{Atten}^i_{mix} + \overline{Atten}^{i-1}_{mix} \qquad (13)$$

Subsequently, a Frequency Operator (FO) $\psi(\cdot)$ is adopted to mixed frequency attention maps: $\psi(\overline{MF_i})$. Given a frequency operator coefficient $\varphi$, where $\varphi \in (0, 1)$, the high-frequency coefficients in $\overline{MF_i}$ are enhanced by $(1 + \varphi)$, making minimal and subtle actions more pronounced. On the other hand, the low-frequency coefficients are reduced by $\varphi$, appropriately diminishing the focus on

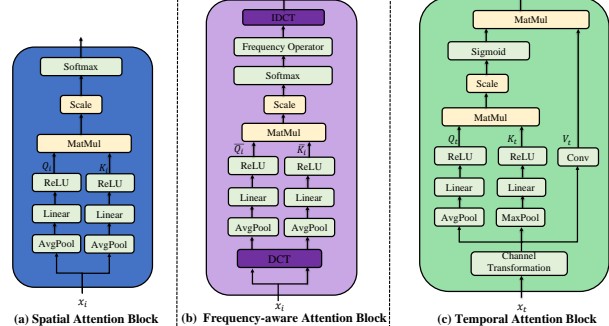

**(a) Spatial Attention Block**    **(b) Frequency-aware Attention Block**    **(c) Temporal Attention Block**

**Figure 3: Three different blocks applied in FreqMixFormer: (a) Spatial Attention Block (SAB), (b) Frequency-aware Attention Block (FAB), and (c) Temporal Attention Block (TAB).**

salient actions while preserving the integrity of overall action representations. The search for the best $\varphi$ is discussed in Section 4.5. Afterward, an IDCT module is employed to restore the transformed skeleton sequence: $MF_i = IDCT(\psi(\overline{MF_i}))$. All the $M_i$ are extracted by Frequency-aware Attention Blocks (FABs), as depicted in Fig. 3 (b). Thus the output is: $MFS_i = MF_i + MS_i$, and the final output of the mixed frequency-spatial attention map can be expressed as:

$$M \leftarrow Concat[MFS_1, MFS_2, \ldots, MFS_i] \qquad (14)$$

We obtain *Value V* from the initial input $X$ with unified computation via adding one spatial $1 \times 1$ convolutional layer along the spatial dimension. Consequently, the input of the Temporal Attention Block is expressed as:

$$x_t = MV \qquad (15)$$

**Temporal Attention Block.** Given the temporal input $x_t$ based on the mixed frequency-spatial attention method, some tricky strategies in [64] are adopted to transform the input channel and acquire more multivariate information alone the temporal dimension: $X_t = CT(x_t)$ (the channel transformation $CT(\cdot)$ is detailed in the Appendix). Then the transformed input $X_t$ is processed with a temporal attention block (Fig. 3 (c)) to obtain the corresponding *Query* and *Key* matrices:

$$Q_t = \sigma(linear(AvgPool(X_t))), \qquad (16)$$
$$K_t = \sigma(linear(MaxPool(X_t))) \qquad (17)$$

And the *Value* in temporal attention block $V_t$ is obtained from the temporal input after a $1 \times 1$ convolutional layer along the temporal dimension. Finally, the temporal attention is expressed as:

$$Atten_{tem} = Softmax(\frac{Q_t K_t^\top}{\sqrt{d}}), \qquad (18)$$

and the final output for the classification head is defined as:

$$X_{out} = (Sigmod(Atten_{tem}))V_t \qquad (19)$$

## 4 Experiments

### 4.1 Datasets

**NTU RGB+D (NTU-60)** [45] is one of the most widely used large-scale datasets for action recognition, containing 56,880 skeleton action samples from 40 subjects across 155 camera viewpoints. Each

**Table 1: Comparison with the SOTA on NTU datasets. \* indicates results are implemented based on the released codes. The highest values are highlighted in red, while the second-highest values are marked in blue.**

| | Method | Source | NTU-60 | | NTU-120 | |
|---|---|---|---|---|---|---|
| | | | X-Sub (%) | X-View (%) | X-Sub (%) | X-Set (%) |
| GCN | ST-GCN [66] | AAAI'18 | 81.5 | 88.3 | 70.7 | 73.2 |
| | AS-GCN [24] | CVPR'19 | 86.8 | 94.2 | 78.3 | 79.8 |
| | 2S-AGCN [46] | CVPR'19 | 88.5 | 95.1 | 82.5 | 84.2 |
| | NAS-GCN [40] | AAAI'20 | 89.4 | 95.7 | - | - |
| | Sym-GNN [25] | TPAMI'22 | 90.1 | 96.4 | - | - |
| | Shift-GCN [9] | CVPR'20 | 90.7 | 96.5 | 85.9 | 87.6 |
| | MS-G3D [34] | CVPR'20 | 91.5 | 96.2 | 86.9 | 88.4 |
| | EfficientGCN-B4 [51] | TPAMI'22 | 91.7 | 95.7 | 88.3 | 89.1 |
| | CTR-GCN [8] | ICCV'21 | 92.4 | 96.8 | 88.9 | 90.6 |
| | FR-Head [75] | CVPR'23 | 92.8 | 96.8 | 89.5 | 90.9 |
| | Koopman [59] | CVPR'23 | 92.9 | 96.8 | 90.0 | 91.3 |
| | Stream-GCN [67] | IJCAI'23 | 92.9 | 96.9 | 89.7 | 91.0 |
| | WDCE-Net [7] | ICASSP'24 | 93.0 | 97.2 | - | - |
| | InfoGCN (4-stream) [10] | CVPR'22 | 92.7 | 96.9 | 89.4 | 90.7 |
| | InfoGCN (6-stream) [10] | CVPR'22 | 93.0 | 97.1 | 89.8 | 91.2 |
| | HD-GCN (4-stream) [22] | ICCV'23 | 93.0 | 97.0 | 89.8 | 91.2 |
| | HD-GCN (6-stream) [22] | ICCV'23 | 93.4 | 97.2 | 90.1 | 91.6 |
| Transformer | ST-TR [41] | CVIU'21 | 90.3 | 96.3 | 85.1 | 87.1 |
| | 4s-GSTN [17] | Symmetry'22 | 91.3 | 96.6 | 86.4 | 88.7 |
| | DSTA [47] | ACCV'20 | 91.5 | 96.4 | 86.6 | 89.0 |
| | STST [70] | ACMMM'21 | 91.9 | 96.8 | - | - |
| | STAR-Transformer [2] | WACV'23 | 92.0 | 96.5 | 90.3 | 92.7 |
| | FG-STFormer [14] | ACCV'22 | 92.6 | 96.7 | 89.0 | 90.6 |
| | SiT-MLP [69] | arXiv'23 | 92.3 | 96.8 | 89.0 | 90.2 |
| | TranSkeleton [31] | TCSVT'23 | 92.8 | 97.0 | 89.4 | 90.5 |
| | Hyperformer [76] | arXiv'22 | 92.9 | 96.5 | 89.9 | 91.3 |
| | STEP-CATFormer [35] | BMVC'23 | 93.2 | 97.3 | 90.0 | 91.2 |
| | SkeMixFormer* (joint only) [64] | ACMMM'23 | 90.7 | 95.9 | 87.1 | 88.9 |
| | SkeMixFormer* (4-stream) [64] | ACMMM'23 | 92.8 | 96.9 | 90.0 | 91.2 |
| | SkeMixFormer* (6-stream) [64] | ACMMM'23 | 93.0 | 97.1 | 90.1 | 91.3 |
| | FreqMixFormer (ours, joint only) | | 91.5 | 96.0 | 87.9 | 89.1 |
| | FreqMixFormer (ours, 4-stream) | | 93.4 | 97.3 | 90.2 | 91.5 |
| | **FreqMixFormer (ours, 6-stream)** | | **93.6** | **97.4** | **90.5** | **91.9** |

3D skeleton data consists of 25 joints. The data is classified into 60 classes with two benchmarks. 1) Cross-Subject (X-Sub): half of the subjects are set for training, and the rest are used for testing. 2) Cross-View (X-View): training and test sets are split based on different camera views (2, 3 views for training, 1 for testing).

**NTU RGB+D 120 (NTU-120)** [32] is an expansion dataset of NTU RGB+D, containing 113,945 samples with 120 action classes performed by 106 subjects. There are two benchmarks. 1) Cross-Subject (X-Sub): 53 actions are used for training, and the rest are used for testing. 2) Cross-Setup (X-Set): samples with even setup IDs are set as training sets, and samples with odd setup IDs are used for testing.

**Northwestern-UCLA (NW-UCLA)** [54] is a 10-classes action recognition dataset containing 1494 video clips. Three Kinect cameras capture the actions with different camera views. We adopt the commonly used evaluation protocols: the first two camera views are used for training, and the testing set comes from the other camera.

### 4.2 Implementation Details

We follow the standard data processing method from [8] to preprocess the skeleton data. The proposed method is implemented on Pytorch [39] with two NVIDIA RTX A6000 GPUs. The model is trained with 100 epochs and 128 batch size for all datasets mentioned above and a warm-up at the first 5 epochs. The weight decay is 0.0005, and the learning rate is initialized to 0.1 in the NTU RGB+D and NTU RGB+D 120 datasets (with a 0.1 reduction at the 35th, 55th, and 75th rounds) and 0.2 in the Northwestern-UCLA dataset (with a 0.1 reduction at the 50th round). A commonly used multi-stream ensemble method [10] is implemented for 4-stream

fusion and 6-stream fusion. The experimental results are shown in Table 1 and Table 2.

### 4.3 Comparison with the State-of-the-Art

In this section, we conduct a comprehensive performance comparison with the state-of-the-art (SOTA) methods on NTU RGB+D, NTU RGB+D 120, and NW-UCLA datasets to demonstrate the competitive ability of our FreqMixFormer. The comparison is made with three ensembles of different modalities and the details are provided in the appendix. Comparisons for NTU datasets are shown in Table 1. We compare our model with the recent SOTA methods based on their frameworks (GCN and Transformer). The recognition accuracy of our FreqMixFormer has outperformed all the transformer-based methods. It is noted that even our 4-stream ensemble results on both NTU-60 (93.4% in X-Sub and 97.3% in X-View) and NTU-120 datasets (90.2% in X-Sub and 91.5% in X-Set) have exceeded all SOTA approaches. Despite the predominant role of GCN-based methods in the skeleton-based action recognition, as we mentioned in Section 1, FreqMixFormer still surpasses the recent methods, such as InfoGCN [10] and HD-GCN [22]. Moreover, our method achieves better performance compared with all the methods that also focus on recognizing discriminative subtle actions, including FR-Head [75] (outperformed by 0.8% on NTU-60 X-Sub and 0.6% on X-View) and WDCE-Net [7] (outperformed by 0.6% on NTU-60 X-Sub and 0.2% on X-View) methods. It is worth noting that our method not only surpasses the existing SOTA GCN-based methods but also enhances the transformer's ability to learn discriminative representations among subtle actions.

In addition to experiments on large-scale datasets like NTU-60 and NTU-120, we extend our research to the small-scale dataset NW-UCLA to further validate our model's performance across different data scales. Table 2 shows results on the NW-UCLA dataset. FreqMixFormer achieves the best results (97.7 %) in comparison to SOTA methods based on GCNs and transformers. Our method outperforms the HD-GCN by 0.8% and SkeMixFormer by 0.3%.

### 4.4 Comparison of Complexity with Other Models

Table 3 shows the complexity comparison with other models. For a fair comparison, we conduct the experiments under the same settings. Although our FreqMixFormer is less efficient (GCNs typically require fewer parameters and incur lower computational costs than Transformers since GCNs leverage the inherent structure of graph data, which allows them to model node dependencies directly with minimal parameters. Additionally, GCN operations are confined to the edges of a graph, significantly reducing the GFLOPs. In contrast, transformers process all pairwise element interactions in a sequence, leading to a rapid increase in computational complexity and parameter count, especially for long sequences) than GCN-based methods, we achieve a very competitive result on the NTU-120 dataset X-Sub, which outperforms HD-GCN by 2.2 % and SKeMixFormer by 0.8 % with fewer parameters than SkeMixformer.

### 4.5 Ablation Study

In this section, we first evaluate the role of key modules in FreqMix-Former, including FAB, FO, and TAB, to analyze the effectiveness of

**Table 2: Comparison with the SOTA on NW-UCLA dataset. The best performance is highlighted in bold. * indicates results are implemented based on the released codes.**

| | Method | Source | NW-UCLA Top-1 (%) |
|---|---|---|---|
| GCN | Shift-GCN [9] | CVPR'20 | 94.6 |
| | CTR-GCN [8] | ICCV'21 | 96.5 |
| | InfoGCN [10] | CVPR'22 | 96.0 |
| | FR-Head [75] | CVPR'23 | 96.8 |
| | Koopman [59] | CVPR'23 | 96.8 |
| | Stream-GCN [67] | IJCAI'23 | 96.8 |
| | HD-GCN [22] | ICCV'23 | 96.9 |
| Transformer | 4s-GSTN [17] | Symmetry'22 | 95.9 |
| | STST [70] | ACMMM'21 | 97.0 |
| | FG-STFormer [14] | ACCV'22 | 97.0 |
| | SiT-MLP [69] | arXiv'23 | 96.5 |
| | Hyperformer [76] | arXiv'22 | 96.7 |
| | SkeMixFormer* [64] | ACMMM'23 | 97.4 |
| | **FreqMixFormer (ours)** | | **97.7** |

**Table 3: Comparison of the complexity of the joint stream state-of-the-art. The best performances are bolded.**

| | Method | NTU-120 X-Sub (%) | Param (M) | GFLOPs |
|---|---|---|---|---|
| GCN | MS-G3D [34] | 84.9 | 3.22 | 5.22 |
| | CTR-GCN [8] | 84.9 | **1.46** | 1.97 |
| | InfoGCN [10] | 85.1 | 1.57 | 1.68 |
| | HD-GCN [22] | 85.7 | 1.68 | **1.60** |
| Transformer | Hyperformer [76] | 87.3 | 2.69 | 9.64 |
| | SkeMixFormer [64] | 87.1 | 2.08 | 2.39 |
| | **FreqMixFormer (ours)** | **87.9** | 2.04 | 2.40 |

each block. Then we propose to search for the best variants within the model, including the number of unit group $n$ for input partition and the best frequency operator coefficient $\varphi$. Additionally, we provide the visualization of the attention maps to show the effectiveness of the mixed frequency-spatial attention mechanism.

**The Design of Frequency-aware Mixed Transformer.** As the results shown in Table 4, the baseline only contains the basic Spatial MixFormer module from [64], which only achieves 89.8% accurate. Then we propose 3 modules for analysis: 1) Frequency-aware Attention Block (FAB): the key part in our proposed method, extracting frequency-based attention maps from the joint sequence, leading to a 1.2% improvement in our baseline. 2) Frequency Operator (FO): an extra module within FABs enhances the high-frequency coefficients and reduces the low-frequency coefficients based on the DCT operator coefficient, resulting in a 1.4% improvement in the baseline. 3) Temporal Attention Block (TAB): a module utilized to learn joint correlations across frames, leading to a 0.9% over the baseline performance. As we can see, each of these modules can enhance the baseline's performance, where the main contribution comes from the utilization of Frequency-aware Attention Blocks with a proper frequency operator coefficient. The best result comes from the combination of all these modules with baseline, which achieves 91.5% accuracy in NTU-60 X-Sub with joint modality. It is speculated that the proposed frequency-aware attention mechanism (see in Section 3.4) plays a significant role in enhancing the action recognition performance. The experiment results on confusing actions with subtle motions are presented in Section 4.6.

**Search for the best frequency operator coefficient $\varphi$.** Moreover, we also conduct analysis regarding the best number of the

**Table 4: The design of Frequency-aware Mixed Transformer.**

| Baseline | FAB | FO | TAB | NTU-60 X-Sub (%) |
|---|---|---|---|---|
| ✓ | ✗ | ✗ | ✗ | 89.8 |
| ✓ | ✗ | ✗ | ✓ | 90.7 (↑ 0.9) |
| ✓ | ✓ | ✗ | ✗ | 91.0 (↑ 1.2) |
| ✓ | ✓ | ✓ | ✗ | 91.2 (↑1.4) |
| ✓ | ✓ | ✗ | ✓ | 91.1 (↑1.3) |
| ✓ | ✓ | ✓ | ✓ | **91.5 (↑ 1.7)** |

unit $n$ in Table 5. It can be seen in Table 5 that the increasing $n$ does improve the results ($n = 2, 3, 4$) with a peak accuracy of 91.5% in NTU-60 X-Sub and 96.0% in NTU-60 X-View. However, further increments do not lead to better outcomes, only to a higher computational cost (the model parameters keep increasing from 1.26M to 2.83M). Given the trade-off between cost and performance, we opt for a splitting number of $n = 4$ for subsequent experiments.

**Search For the Best Frequency Operator Coefficient $\varphi$.** In Table 6, we investigate the impact of frequency operator coefficient $\varphi$. As we discussed on Section 3.4, the high-frequency coefficients will be amplified by $(1 + \varphi)$, and the low-frequency coefficients will be diminished by $\varphi$. As $\varphi$ increases from 0.1 to 0.5, there is a general trend of improved performance, reaching a peak at $\varphi = 0.5$, which achieves the highest accuracy of 91.5% on NTU-60 X-Sub and 96.0% on X-View. However, further increasing $\varphi$ from 0.6 to 0.9 does not lead to improvements in performance. In fact, the accuracy slightly declines. This suggests that enhancing high-frequency components too much or reducing low-frequency components too aggressively may lead to loss of motion patterns learning.

**Effectiveness of the Mixed Frequency-aware Attention.** Fig. 4 presents the visualization of the attention matrices learned by FreqMixFormer. The skeleton configuration is generated from the NTU-60 dataset (Fig. 4 (a)). We take "eat meal" as an example (Fig. 4 (b)). In the correlation matrix, a more saturated yellow represents a large weight, indicating a stronger correlation among joints. And the numbers denote different joints. Note that the Mixed Spatial Attention Map (Fig. 4 (c), learned by SAB) represents the spatial relationships among joints. The Mixed Frequency Attention Map (Fig. 4 (d), learned by FAB) suggests the frequency aspects of motion. Based on these two attention maps, a mixed frequency-spatial attention map is proposed (Fig. 4 (e)) for capturing both spatial correlations and frequency dependencies, integrating the spatial and frequency skeleton features.

**Table 5: Search for the best number of the unit $n$.**

| | NTU-60 | | |
|---|---|---|---|
| $n$ | X-Sub (%) | X-View (%) | Param (M) |
| 2 | 90.0 | 95.1 | 1.26 |
| 3 | 90.8 | 95.3 | 1.64 |
| 4 | **91.5** | **96.0** | 2.04 |
| 5 | 91.3 | 95.9 | 2.45 |
| 6 | 91.3 | 95.7 | 2.83 |

**Table 6: Search for the best frequency operator coefficient $\varphi$.**

| | NTU-60 | |
|---|---|---|
| $\varphi$ | X-Sub (%) | X-View (%) |
| 0.1 | 90.9 | 95.6 |
| 0.2 | 90.8 | 95.4 |
| 0.3 | 91.0 | 95.6 |
| 0.4 | 90.9 | 95.7 |
| 0.5 | **91.5** | **96.0** |
| 0.6 | 91.0 | 95.8 |
| 0.7 | 91.0 | 95.6 |
| 0.8 | 91.0 | 95.7 |
| 0.9 | 91.1 | 95.6 |

As we see in the figures, the model focuses on the correlations with the spine and right-hand tip in the spatial domain. As for the frequency domain, more correlation areas are concerned (joint connections with the spine, left arm, and the interactions between head

and hands), which indicates the model is analyzing more discriminative movements overlooked in the spatial domain. Meanwhile, the mixed frequency-spatial attention map contains not only the strong attention areas learned from spatial space but also the concerned correlations in frequency space. This demonstrates that our FreqMixFormer model advances this by extracting minimal and subtle joint representations (highlighted with the red box in Fig. 4 (b)) from both spatial and frequency domains. The effectiveness of the mixed frequency attention is also verified in Table 4.

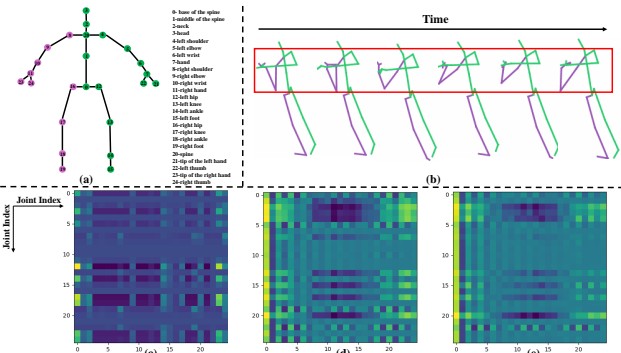

**Figure 4: The visualization of attention matrices. (a) is the joint index of the NTU RGB+D dataset. (b) is the skeleton sequence of the "eat meal" action. The red box indicates the deeper attention area among joints. (c) is the mixed spatial attention map extracted from the spatial attention block. (d) is the mixed frequency attention map extracted from the frequency-aware attention block. (e) is the mixed frequency-spatial attention map, representing the mixed frequency-spatial skeleton features.**

## 4.6 Comparison Results on Confusing Actions

To validate our model's capability in discerning discriminative actions, similar to [7, 75], we categorize certain actions from the NTU-60 dataset (only joint stream with X-Sub protocol) into three sets based on the classification results of Hyperformer [76]: the actions with accuracy lower than 80% as Hard set, between 80% and 90% as Medium set, and higher than 90% as Easy set. All the confusing actions are classified into Hard and Medium sets. For example, "writing," "reading," and "playing with a phone" are categorized as Hard action sets due to their subtle differences, which are limited to small upper-body movements involving only a few joint correlations, leading to low recognition results. We compare our results with the recent transformer-based models Hyperformer [76] and SkeMixFormer [64]. The results of different difficult-level actions are displayed in Table 7, showcasing that our model outperforms the recent SOTA methods across these three subsets. Furthermore, the detailed results of Hard and Medium actions are also provided in Fig. 5 and Fig. 6. The results indicate that our method significantly enhances performance on both hard-level and medium-level confusing actions, demonstrating its capability to differentiate ambiguous movements.

**Table 7: The results on different difficult-level actions for NTU RGB+D dataset.**

| Method | NTU-60 X-Sub (%) | | |
|---|---|---|---|
| | **Hard** | **Mdeium** | **Easy** |
| Hyperformer [76] | 71.4 | 83.6 | 94.1 |
| SkeMixFormer [64] | 71.9 | 84.6 | 94.3 |
| **FreqMixFormer (ours)** | **73.9** | **86.1** | **95.2** |

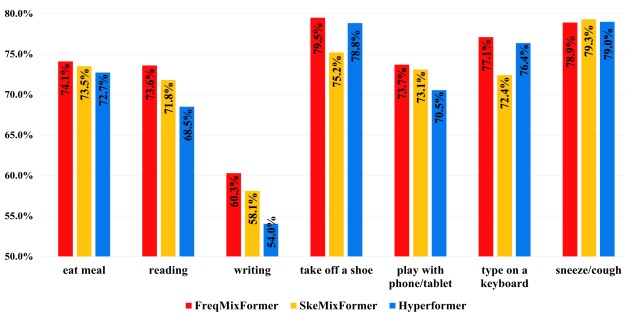

**Figure 5: Accuracy comparison results on confusing actions in the hard set.**

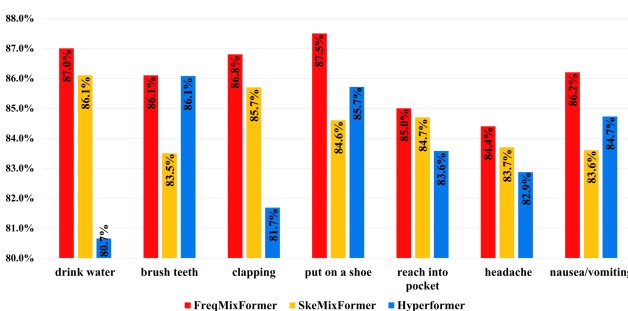

**Figure 6: Accuracy comparison results on confusing actions in the medium set.**

## 5 Conclusion and Discussion

In this work, we introduce Frequency-aware Mixed Transformer (FreqMixFormer), a novel transformer architecture designed to discern discriminative movements among similar skeletal actions by leveraging a frequency-aware attention mechanism. This model enhances skeleton action recognition by integrating spatial and frequency features to capture comprehensive intra-class frequency-spatial patterns. Our extensive experiments across diverse datasets, including NTU RGB+D, NTU RGB+D 120, and NW-UCLA, establish FreqMixFormer's state-of-the-art performance. The proposed model demonstrates superior accuracy in general and significant advancements in recognizing confusing actions. Our research advances the field by presenting a method that integrates frequency domain analysis with current transformer models, paving the way for more precise and efficient action recognition systems. This work is anticipated to inspire future research on precision-targeted skeletal action recognition.

## Acknowledgment

This work was supported in part by the NSF grant of 1840080.

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
