# OpenReview forum: "Frequency Guidance Matters: Skeletal Action Recognition by Frequency-Aware Mixed Transformer"
_acmmm.org/ACMMM/2024/Conference — MM2024 Poster_

### Official Review · Reviewer_mtVv · 2024-05-20

**Rating:** 5
**Confidence:** 4

**Summary:**

The paper "Frequency Guidance Matters: Skeletal Action Recognition by Frequency-Aware Mixed Transformer" presents a groundbreaking approach in the field of skeleton action recognition. The authors introduce the Frequency-aware Mixed Transformer (FreqMixFormer), a model designed to capture the subtleties of human motion through a novel integration of spatial and frequency features. By leveraging the power of Discrete Cosine Transform (DCT), the FreqMixFormer embeds joint features into frequency attention maps, allowing for the discrimination of movements based on their frequency coefficients. This innovative method results in a significant enhancement in recognizing actions that are similar yet distinct, a challenge often faced in computer vision tasks.

The architecture of the FreqMixFormer is thoughtfully crafted, with a Frequency-aware Attention module that is pivotal in transferring spatial joint information to the frequency domain. This is further complemented by a mixed transformer architecture that fuses spatial features with frequency features, and a temporal transformer that extracts global correlations across frames. The result is a comprehensive model that not only outperforms existing state-of-the-art methods on popular datasets such as NTU RGB+D, NTU RGB+D 120, and NW-UCLA but does so with an elegant efficiency that is notable in the field of transformers.

**Strengths:**

1. The authors provide a meticulous ablation study that dissects the contributions of key modules like the Frequency-aware Attention Block (FAB), Frequency Operator (FO), and Temporal Attention Block (TAB). This study not only validates the effectiveness of each component but also underscores the collective impact they have on improving the baseline's performance. The complexity analysis presented in the paper is particularly insightful, offering a comparison that highlights the FreqMixFormer's competitive edge over other models in terms of parameter count and computational efficiency.
2. One of the standout aspects of the paper is the visualization of attention matrices, which provides a vivid demonstration of the model's focus on discriminative joint correlations across both spatial and frequency domains. This visual evidence solidifies the FreqMixFormer's capability to learn from subtle motion features, setting it apart from traditional models that may overlook such nuances.
3. Moreover, the FreqMixFormer's performance on actions that are typically challenging to distinguish is particularly impressive. The model's ability to excel in recognizing even the most confusing actions with subtle differences speaks to its robustness and the potential for its application in real-world scenarios where precision is paramount.

**Limitations:**

1. The Frequency Operator (FO) makes the smallest and subtle movements more noticeable. Can the author provide more explanation or datagram to prove this point?
2. I noticed that DCT and IDCT are important components, but the authors did not pay much attention to them in ablation experiments. Hopefully the author will add this part.
3. I am interested in the design of MFi matrices. It looks like concatenation is done in the channel dimension, but matrix addition is done in the formula. Can the author be further clarified? Also, if addition is used, is matrix multiplication considered because it can mix low-frequency with high-frequency features.
4. Skeleton MixFormer uses Channel Reforming to smooth out the feature splitting caused by channel grouping, which I noticed was not addressed in the paper. How did the author consider or see the benefits of not adopting Channel Reforming?
5. The authors need to add a supplementary note of the computational cost in Table 4.

**Suitability:**

3

---

### Official Review · Reviewer_ckag · 2024-05-25

**Rating:** 3
**Confidence:** 4

**Summary:**

In this paper, the authors observe that while current transformer-based methods demonstrate substantial potential in modeling long-term dependencies in skeleton sequences, they fall short in extracting discriminative representations between subtly distinct actions with similar kinetic patterns. To address this problem, the paper proposes the "Frequency-aware Mixed Transformer (FreqMixFormer)" model, tailored to address this challenge.

The novelty of this research centers on the development of a Frequency-Aware Attention Block (FAB), which decouples skeletal frequency representations by embedding joint features into frequency attention maps. This strategy aims to differentiate discriminative movements based on their frequency coefficients. Moreover, a mixed transformer architecture is devised to integrate spatial and frequency features, comprehensively modeling frequency-spatial patterns within action classes. A complementary Temporal Attention Block (TAB) is proposed to capture global inter-frame joint correlations. Extensive experimentation evidences the superiority of FreqMixFormer over state-of-the-art (SOTA) performance on three prominent skeletal action recognition datasets, namely NTU RGB+D, NTU RGB+D 120, and NW-UCLA, highlighting its advanced general accuracy and remarkable progress in deciphering confusing actions.

**Strengths:**

1. Enhanced Action Feature Discrimination via Frequency Domain: The authors decompose actions within skeletal sequences according to distinct frequency components and innovatively employ a Frequency-Aware Attention Block (FAB) for feature extraction, allowing targeted amplification or attenuation of features at varying frequencies. Experimental outcomes confirm the efficacy of this approach in significantly improving recognition accuracy for actions with subtle differences.

2. Integration of Spatial-Temporal Features: The authors effectively merge both spatial and temporal features through the designed FreqMixFormer architecture, facilitating efficient learning of inter-frame joint correlations in the temporal dimension and action features in the spatial domain. This leads to enhanced performance across multiple datasets.

3. Optimized Model Complexity: Relative to state-of-the-art (SOTA) models, FreqMixFormer achieves high accuracy while maintaining a reasonable parameter count and computational complexity. Specifically, on the NTU-120 dataset, it attains 87.9% accuracy with 2.04 million parameters and a computational cost of 2.40 GFLOPs, manifesting an advantageous balance between algorithmic efficiency and performance capabilities.

**Limitations:**

1. According this paper, its contributions are unclear. As is widely known, frequency typically conveys the overall information of a signal. Skeleton data can be regarded as a unique kind of signal.  In my perspective, the frequency of skeleton data emphasizes global motion patterns, whereas discriminative features are largely extracted from local joint points. Might it be beneficial to employ Discrete Wavelet Transform (DWT) instead of Discrete Cosine Transform (DCT) for obtaining the frequency domain representation of skeleton data?

2. In the article, MS stands for Mixed Spatial Attention Map, while MF denotes Mixed Frequency Attention Map. Although MF undergoes an Inverse Discrete Cosine Transform (IDCT) operation to transform back to the time domain, in terms of features, they still belong to distinct domains. However, when fusing these two types of features to obtain the Mixed Frequency Spatial Attention Block, why is the summation operation solely adopted?

3. In Figure 2, the operation of the Frequency Operator (FO) is not illustrated. Moreover, the partitioning of the embedded data (X) in the figure contains duplicate elements, leading to an unclear explanation.

4. Figure 4 showcases the visualizations of attention maps extracted by different modules alongside brief explanations. Nonetheless, the figure lacks an indication of the magnitude corresponding to different colors, and the depictions in sub-figures (d) and (e) regarding the attention maps' correlation with joint actions remain inadequately defined.

**Suitability:**

2

---

### Official Review · Reviewer_hcdT · 2024-05-25

**Rating:** 4
**Confidence:** 3

**Summary:**

This manuscript introduces a novel skeletal action recognition method named FreqMixFormer. The approach is designed to identify and differentiate similar human skeletal movements that have subtle differences by employing a specialized frequency-aware mechanism. The FreqMixFormer model uses the Discrete Cosine Transform to translate the sequence data of skeletal actions into the frequency domain, where it captures features through attention mechanisms. Through extensive experiments on several well-known datasets, the effectiveness of FreqMixFormer in the task of skeletal action recognition has been proven.

**Strengths:**

The manuscript demonstrates a high degree of clarity in describing the theoretical foundations, methodological design, experimental setup, and result analysis. Illustrations and tables assist in elucidating complex concepts, enabling readers to more easily comprehend the content of the paper. Additionally, the successful application of FreqMixFormer in the task of skeletal action recognition indicates that it can be applied to a variety of scenarios, possessing significant practical application value.

**Limitations:**

1, In Section 3.2, the author states "where 𝑋 …… to facilitate interpretable learning of joint adjacency." I am unclear as to how the grouping strategy enhances interpretability, and I kindly ask the author to provide further explanation.

2, In Figure 2, the authors present their network architecture. Typically, transformer-based networks consist of multiple layers of blocks. However, I did not see any discussion about the number of encoder layers in the ablation study section. The authors should provide more explanation about this.

3, In Table 3, the authors provide the number of parameters and computational complexity. However, I believe the article’s persuasiveness could be further enhanced if the authors offered additional information, such as the GPU memory usage during training compared to the baseline, as well as the time required for training and inference.

4, In Table 4, the authors mention "fo," but its detailed design and the rationale for its parameter selection are not entirely clear to me. If the authors could provide more specifics, it would greatly enhance the reader's understanding.

5, In Section 4.6, the authors utilize different subsets of data with varying levels of difficulty. Since the division between easy and difficult examples is based on the accuracy of Hyperformer, is it still fair to include it in the comparison list? Why not include more comparative methods in Table 7? For instance, GCN-based approaches.

**Suitability:**

3

---

### Meta-Review · Area_Chair_Ny7a · 2024-07-05

**Recommendation:** Accept (Poster)
**Confidence:** 4

**Metareview:**

This paper develops a frequency-aware mixed Transformer for recognizing similar skeletal actions with subtle discriminative motions. Experiments on three datasets demonstrate the effectiveness of the proposed method. After rebuttal, two reviewers give positive scores. AC thinks that the questions of all the reviewers have been answered well by the authors in the rebuttal, and so recommends to accept.

Quality: The paper is well structured and easy to read.

Clarity: The paper is clearly presented.

Originality: The proposed method is interesting and effective.

Significance: FreqMixFormer can be applied to a variety of scenarios, which has significant practical application value.